# Novel Susceptibility Genes Drive Familial Non-Medullary Thyroid Cancer in a Large Consanguineous Kindred

**DOI:** 10.3390/ijms24098233

**Published:** 2023-05-04

**Authors:** Pierre Majdalani, Uri Yoel, Tayseer Nasasra, Merav Fraenkel, Alon Haim, Neta Loewenthal, Raz Zarivach, Eli Hershkovitz, Ruti Parvari

**Affiliations:** 1The Shraga Segal Department of Microbiology, Immunology & Genetics, Faculty of Health Sciences, Ben-Gurion University of the Negev, Beer-Sheva 84105, Israel; pierrem@post.bgu.ac.il; 2The National Institute for Biotechnology in the Negev, Ben-Gurion University of the Negev, Beer-Sheva 84105, Israel; 3Endocrinology Unit, Soroka University Medical Center and Faculty of Health Sciences, Ben-Gurion University of the Negev, Beer-Sheva 84101, Israel; uriy@bgu.ac.il (U.Y.);; 4Internal Medicine A, Soroka University Medical Center and Faculty of Health Sciences, Ben-Gurion University of the Negev, Beer-Sheva 84101, Israel; 5Pediatric Endocrinology Unit, Soroka University Medical Center and Faculty of Health Sciences, Ben-Gurion University of the Negev, Beer-Sheva 84101, Israelelih@clalit.org.il (E.H.); 6Department of Life Sciences, Faculty of Natural Sciences, Ben-Gurion University of the Negev, Beer-Sheva 84101, Israel

**Keywords:** familial non-medullary thyroid cancer (FNMTC), papillary thyroid carcinoma (PTC), whole exome sequence (WES), protein–protein interactions (PPIs), *ARHGEF28*, *FBXW10*, *SLC47A1*, *XRCC1*, *HRAS*

## Abstract

Familial non-medullary thyroid cancer (FNMTC) is a well-differentiated thyroid cancer (DTC) of follicular cell origin in two or more first-degree relatives. Patients typically demonstrate an autosomal dominant inheritance pattern with incomplete penetrance. While known genes and chromosomal loci account for some FNMTC, the molecular basis for most FNMTC remains elusive. To identify the variation(s) causing FNMTC in an extended consanguineous family consisting of 16 papillary thyroid carcinoma (PTC) cases, we performed whole exome sequence (WES) analysis of six family patients. We demonstrated an association of *ARHGEF28*, *FBXW10*, and *SLC47A1* genes with FNMTC. The variations in these genes may affect the structures of their encoded proteins and, thus, their function. The most promising causative gene is *ARHGEF28,* which has high expression in the thyroid, and its protein-protein interactions (PPIs) suggest predisposition of PTC through ARHGEF28-SQSTM1-TP53 or ARHGEF28-PTCSC2-FOXE1-TP53 associations. Using DNA from a patient’s thyroid malignant tissue, we analyzed the possible cooperation of somatic variations with these genes. We revealed two somatic heterozygote variations in *XRCC1* and *HRAS* genes known to implicate thyroid cancer. Thus, the predisposition by the germline variations and a second hit by somatic variations could lead to the progression to PTC.

## 1. Background

Differentiated thyroid cancer (DTC), the most common endocrine cancer, is derived from follicular cells, as opposed to the rarer medullary thyroid cancer (MTC) of thyroid C-cell origin. DTC has three histological subtypes: papillary thyroid carcinoma (PTC), follicular thyroid carcinoma (FTC), and Hürthle-cell carcinoma, accounting for at least 80% of the total and approximately 90% of all new cases [1,2]. According to the Surveillance, Epidemiology, and End Results (SEER) database (https://seer.cancer.gov/statfacts/html/thyro.html, accessed on 7 July 2022), its yearly incidence in the United States is 14 new cases per 100,000 persons with approximately 3:1 female-to-male ratio, with the lifetime risk of developing DTC being ~1.2%. Family history is a well-known risk factor for DTC [3,4]. Using the Swedish Cancer Registry from 1958 to 2002, it was found that the standardized incidence ratio (the ratio of the observed to the expected number of cases) for PTC was 3.21 and 6.24 when a parent and a sibling, respectively, were diagnosed previously with DTC [5]. Comparable results were presented more recently in a nationwide cohort study from Taiwan [6].

DTC in two or more first-degree relatives without other predisposing hereditary or environmental causes defines familial non-medullary thyroid cancer (FNMTC) [7]. FNMTC accounts for 3.2–9.6% (and in some surveys may reach 15%) of all DTC cases typically presented with an autosomal dominant pattern of inheritance with incomplete penetrance [8,9]. As with sporadic cases of DTC, PTC is the most prevalent histological subtype, and the female-to-male ratio is 3:1 [10,11]. There are some uncertainties regarding the clinical presentation of FNMTC. Most studies demonstrate younger age at presentation, multifocality, and more aggressive histopathological features compared to sporadic DTC [7,8,12,13,14].

In families with three to five affected FNMTC members, the likelihood of a familial trait is 96% [15]. Genetic evaluation is complex because family members of FNMTC patients are at increased risk of developing benign thyroid diseases, including follicular adenoma, Hashimoto’s thyroiditis, and multinodular goiter (MNG) [6]. This variable expression of FNMTC suggests that the responsible gene(s) may lead to predisposition or susceptibility to thyroid cancer and not directly to cancer development.

Several chromosomal loci (1p13.2–1q22, 2q21, 14q32, 19p13.2, and 8p23.1–p22) [16,17,18,19] and low-penetrance mutations in *NKX2.1, FOXE1, HABP2, RTFC, MYH9,* and *CHEK2* have been identified to confer susceptibility to non-syndromic FNMTC [20,21,22,23,24,25]. *MYO1F* and *DICER1* were proposed to be the causative genes in 19p13.2 and the 14q32 loci, respectively [26,27,28,29]. Recent data have also shown that dysregulation of miRNA expression is a hallmark of thyroid cancer [30]. Predisposing risk variants were found in non-coding genes, including miRNAs [31] and a long non-coding RNA gene *PTCSC2* [32]. Altered splicing regulation has been reported in FNMTC patients carrying a germline mutation in the *SRRM2* gene, encoding a splicing machinery subunit [33]. Germline mutations were also identified as activating tumorigenic signaling pathways in *MAP2K5* [34] and *SPRY4* [34,35] and in genes encoding RAS pathway regulators, such as RASAL1 and SRGAP1 [36,37]. Studies conducted in patients with DTC and FNMTC identified germline mutations in DNA repair genes (e.g., BRCA1/2, ATM, and XRCC1) [38,39,40,41]. These data indicate that the genetic predisposition to FNMTC is characterized by a high degree of heterogeneity, and the genetic and molecular basis for most FNMTC remains elusive.

Herein we report the results of a detailed clinical evaluation and exome sequencing analysis of a large consanguineous Bedouin family, with many cases of PTC and MNG, to identify the germline variations that may generate the susceptibility for their neoplastic thyroid diseases.

## 2. Results

### 2.1. Clinical Findings

The proband (III8) was evaluated at the endocrine unit of Soroka University Medical Center (SUMC) following thyroidectomy, with a final pathology report of PTC. During this index visit, he reported that three siblings and a few other second-degree relatives were diagnosed with PTC. Accordingly, we offered full clinical assessment for all family members of the large Bedouin kindred. The clinical assessment included thorough medical history, physical examination, biochemical evaluation, neck ultrasound (US), and thyroid US-guided fine needle aspiration (FNA) for cytological biopsy evaluation when thyroid nodules were found.

Fourteen patients with PTC or a multinodular goiter (MNG) in the kindred were evaluated. We mainly focused on the two left branches (according to the pedigree, Figure 1), which showed the most significant number of family members with thyroid pathology (Table 1). In the third generation (III in the pedigree) of the left branch, four out of ten siblings were diagnosed with PTC and two others with benign MNG, while three had a normal thyroid gland per the US. In the second branch from the left, three out of five siblings of generation III had PTC. In addition, they reported that their mother died of thyroid cancer at a young age (objective data are not available). According to our knowledge, only one patient in generation IV was diagnosed with PTC. Of the eight patients with documented PTC, four were males and four were females. The age at the time of PTC diagnosis ranged from 22 to 46. All five patients with available full pathology descriptions had multifocal PTC; in four of them, the histology reported a follicular variant of PTC. At least five patients had clinically significant lymph node metastasis, and seven out of eight were treated with radioactive iodine (at least six with high activity) following thyroidectomy. Despite relatively aggressive disease at presentation, the response to treatment was favorable in all patients, and all patients were free of disease at the last follow-up.

### 2.2. Evaluating and Prioritizing Potential Candidate Variants

As described in the clinical findings, we recruited members of a large Bedouin family, providing a Lod Score of 2.8 for dominant incomplete penetrance (90%) for the two left branches (Figure 1). Both recessive and dominant patterns are feasible because of the high consanguinity of the family. For the recessive mode, an affected person (the mother (II2) of patients XFIII1, XFIII4, and XFIII5 was reported to have died of thyroid cancer) could have siblings with a heterozygous mate, thus showing a pseudo-dominant pattern. A healthy parent could have the mutation without presenting a disease-related phenotype due to incomplete penetrance for the dominant model. In the case of dominant inheritance, all patients should present the mutation in heterozygosity. Still, healthy individuals will not negate a putative-causing mutation because they could be too young to show the FNMTC or non-penetrant. Similarly, those with MNG may have the mutation, but FNMTC has not been developed yet, or the mutation is not causing MNG. Thus, for the patients with MNG, we considered their clinical status regarding FNMTC as unknown, and they were not used for approval or negation in the segregation analysis.

A whole exome sequence (WES) analysis was performed on six patients (Figure 1). No mutations in the genes previously associated with FNMTC [17,20,25,33,35,42,43,44,45,46] were present (Appendix A). We also verified and negated mutations in the transcription factors *PAX8* and *HHEX,* which are crucial for thyroid morphogenesis during embryogenesis and maintaining normal thyroid architecture, differentiation, and function [27].

We employed the Omicia, Franklin, and Ingenuity pipelines to search for shared homozygous or heterozygous variants in the exomes of five patients (III8, III10, IVA5, XFIII4, and XFIII5). We found the three pipelines’ usage necessary since their predictions and filtration tools differ. We looked for shared variations between the five patients of the two left branches: III8, III10, IVA5, XFIII4, and XFIII5 (Figure 1). We used several negation criteria to prioritize the possible candidate variants, detailed in Table 2. After the filtration process, the segregation of the 54 remaining variants was verified by PCR amplification of the DNA containing the variation, followed by Sanger sequencing. Fifty variants were negated by the segregation in the family (Appendix A). Assuming an autosomal dominant pattern of inheritance with incomplete penetrance [8,47], four variants segregated as expected, all with incomplete penetrance (Table 3).

The thyroid expression (according to GTEx Portal), prediction for damage, and frequency in the population of origin for the four remaining candidate variants are detailed in Table 4. The splice region variant in Solute Carrier Family 24 Member 4 (*SLC24A4*, Gene ID: 10978) is less probable for causing FNMTC since its prediction for altering the splice site is benign by splicing prediction models (dbscSNV Ada: <0.01 and splice AI: 0.17) and by the ACMG guidelines (Table 4) [48]. This leaves three potential candidate variants presenting as heterozygous (Table 4): (1) an in-frame deletion in F-Box And WD Repeat Domain Containing 10 (*FBXW10*, Gene ID: 1211), (2) the missense variant in Solute Carrier Family 47 Member 1 (*SLC47A1*, Gene ID: 25588), and (3) the missense variant in Rho Guanine Nucleotide Exchange Factor 28 gene (*ARHGEF28*, Gene ID: 30322).

### 2.3. Evolutionary Conservation and the Effects of the Amino Acid Changes on Protein Conformation

To understand the effect of the amino acid alterations, we used the relevant alpha fold models and analyzed them using PyMOL software. ARHGEF28 Asn108 is not evolutionarily conserved (Figure 2A). To verify the effect of the variation on the protein structure, we used the human ARHGEF28 alpha fold model (accession code—Q8N1W1). The domain surrounding the variation has a confidence level of 70 < pLDDT < 90 (Per-residue Confident Score). The amino acid Asn108 is predicted to locate at the tip of an alpha helix (part B1 of Figure 2). This residue stabilizes the domain fold via a network of hydrogen bonds (dashed lines); thus, the Asn108Ser amino acid change will most likely alter this hydrogen bond network. FBXW10 Ile440 is conserved evolutionarily down to placental mammals (Figure 2A). No further orthologs were found in the National Center for Biotechnology Information database (NCBI). The domain surrounding the variation, predicted by the human FBXW10 alpha fold model (accession code—Q5XX13), has a very high confidence level of pLDDT > 90. The amino acid Ile440 (part B2 of Figure 2) is located in the beta-hairpin of one of the propeller blades. Deleting the Ile440 amino acid will most likely cause the alteration of the beta-hairpin due to possible disruption of the surrounding hydrogen bond network (dashed lines).

SLC47A1 Gly288 is conserved evolutionarily down to fish (Figure 2A). The domain surrounding the variation, predicted by the human SLC47A1 alpha fold model (accession code—Q96FL8), has a very high confidence level of pLDDT > 90. The amino acid Gly288 (part B3 of Figure 2) is located between two alpha-helices. This residue most likely allows the formation of a turn during the transporter folding. Thus, the change of glycine by serine may prevent the needed flexibility. Glycine is known as a helix-deforming residue, and its alteration may cause an extended helix during protein folding.

### 2.4. Analysis of Protein Interactions

We looked for protein–protein interactions (PPIs) that could reveal a biological network between the susceptible variations in the three genes *ARHGEF28*, *FBXW10,* and *SLC47A1* and genes previously related to FNMTC. Using the BioGrid database, we found a significant PPI only for the *ARHGEF28* gene, with Sequestosome 1 *(SQSTM1)* gene detected by affinity chromatography technology [51]. In turn, the *SQSTM1* gene had a PPI with the Tumor Protein P53 (*TP53*) gene, detected by affinity capture (identified by Western blot) and reconstituted complex (an interaction detected between purified proteins in vitro) [52,53,54,55,56]. *TP53* is among the genes which show many genetic alterations in excised malignant thyroid nodules [57] (Appendix A).

In addition, another high-interaction-weight PPI of the *ARHGEF28* gene was with Myosin Heavy Chain 9 (*MYH9*) gene, identified by cross-linking mass spectrometry [58]. MYH9 was previously suggested as a germline genetic risk factor for the development of non-medullary thyroid cancer [25]. MYH9 binds to lncRNA gene *PTCSC2* (Papillary Thyroid Carcinoma Susceptibility Candidate 2) and regulates *FOXE1* (Forkhead box protein E1) in the 9q22 thyroid cancer risk locus [59] (Appendix A).

### 2.5. Evaluating Somatic Candidate Variants

We hypothesized that the research’s germline variation serves as susceptibility and another variation should occur for the malignant progression. To find the somatic variations, we performed WES analysis for the DNA purified from the malignant thyroid tissue of patient III8 (Figure 1). Assuming a two-hit hypothesis that requires both alleles to be inactivated in tumor suppressor genes, we looked for additional variations in *ARHGEF28, FBXW10,* and *SLC47A1* genes. We found no homozygote, heterozygote, or compound heterozygote second-hit variation in these genes. Next, we searched for possible causative variations in other genes. We applied the Franklin pipeline to search for homozygous or heterozygous variants in the exome sequences of the malignant tissue that do not appear in the gDNA exome sequence of the same patient. We looked for alleles with less than 5% frequency in the public databases (gnomAD browser, 1000 Genomes, ExAC, and EVS) in the malignant thyroid exome sequence (see Table 5 for negation criteria). We identified two variations in genes in which germline variations cause susceptibility to FNMTC (Appendix A) and in 13 somatic genes with genetic alterations in malignant thyroid nodules (*NRAS, HRAS, KRAS, THADA, PIK3CA, BRAF, TERT, PAX/PPARG, PTEN, DICER, E1F1AX, TSHR,* and *TP53*) [57].

The two remaining variations were as follows:A heterozygote missense variation in X-Ray Repair Cross Complementing 1 gene (*XRCC1*, Gene ID: 12828) on chromosome 19:44,047,825 (GRCh37/hg19), c.1727A > C (NM_006297.2), p.Asn576Thr. This gene was reported to have mutations causing FNMTC [38]. Patient IVA5 was the only family member to present heterozygosity of the variation in his gDNA sequence. This variant was present in approximately 4% of our collection of Bedouin exomes. No clinical evidence was found for this variant, and it is classified as a benign variant (BS1 and BS2) according to ACMG [48].A heterozygote missense variation in *HRAS* proto-oncogene (Gene ID: 5173) on chromosome 11:533,875 (GRCh37/hg19), c.181C > A (NM_005343.4), p.Gln61Lys. It is one of the 13 genes with genetic alterations in excised malignant thyroid nodules [57]. None of the family patients presented the variation in their gDNA sequence. Moreover, the variation was absent in the public databases (gnomAD browser, 1000 Genomes, ExAC, and EVS), our collection of Bedouin exomes, the database of 77 Bedouins exomes [49], and the Qatari genome of more than 1000 exomes, with a majority of the Bedouin population [50]. According to ACMG, *HRAS* gene variation c.181C > A is classified as a pathogenic variant (PM1, PP2, PM2, PM5, and PP5). The variation is linked to DTC, specifically to FTC, and the follicular variant of PTC, the prominent histopathological variant among the PTC patients in the described family [60,61]. In addition, it was classified as a pathogenic variant in ClinVar and UniProt classification.

## 3. Discussion

To identify the genetic causes of FNMTC, we recruited an extended Bedouin family (Figure 1) with ten family members diagnosed with PTC and four with MNG. We focused mainly on two branches in the family pedigree presenting a similar pattern of inheritance and consisting of the most significant number of family members with PTC and MNG (Table 1). Using WES analysis, we did not identify a promising homozygote candidate variant. Assuming an autosomal dominant inheritance pattern with incomplete penetrance [8,47], we identified four heterozygote potential candidate variants which segregated as expected in the family. The splice region variant in the *SLC24A4* gene had benign splice-altering scores according to different tools. Thus, it can probably be considered a chance variation, not contributing to the causation of FNMTC. The variants in *FBXW10* and *SLC47A1* genes were a second priority because the expression of both genes is very low in normal thyroid tissue. However, they cannot be negated since the expression level could change in thyroid malignancy, and the variations appear to affect the protein’s structure and function. The *FBXW10* gene is a member of the F-box protein family that acts as a protein-ubiquitin ligase. The *SLC47A1* gene encodes a carrier protein of unknown function. Among its related pathways are the transport of glucose and other sugars, bile salts and organic acids, metal ions and amine compounds, and glucose/energy metabolism. PPI analysis could not link *FBXW10* or *SLC47A1* genes to a pathway that contributes to thyroid disorders or malignancies.

Thus, the most promising candidate for the FNMTC causative variation is in the *ARHGEF28* gene. The missense heterozygote variation c.323A > G causes the amino acid change in position 108 from asparagine to serine, with damaging prediction scores of SIFT 0.14 (tolerate prediction), CADD > 10 (10% most deleterious), and PolyPhen 0.02 (benign). The *ARHGEF28* gene is highly expressed in the normal thyroid tissue sample, and the variant found in the gene is extremely rare.

The *ARHGEF28* gene was the only candidate gene with a subnetwork that connects it with genes associated with PTC [57] (Appendix A). The most noticeable was the PPI between *ARHGEF28*-*SQSTM1-TP53* genes. Several findings indicate that alterations in the *p53* sequence play a role in the early stages of thyroid carcinogenesis. Indeed, *p53* mutations were recently found in up to 40% of PTCs and 22% of oncocytic follicular thyroid carcinomas [62]. However, we did not detect any variation in *p53* by the WES analysis of the malignant tissue of patient III8. Another association of the *ARHGEF28* gene with PTC was through the PPI of *ARHGEF28*-*MYH9* genes. The *MYH9* gene binds to the lncRNA gene, *PTCSC2*, and suppresses the expression of *FOXE1*, which regulates the p53 pathway in thyroid cells [59]. Multiple GWAS and familial studies, including functional analyses, strongly support the involvement of *FOXE1* variations in FNMTC etiology [21,63,64].

Our WES analysis of the DNA extracted from the malignant thyroid tissue of patient III8 identified somatic heterozygote missense candidate variations in two genes: *XRCC1* and *HRAS,* reported to have a role in thyroid cancer [38,57]. The *XRCC1* gene encodes a protein with an essential role in the base excision repair (BER) pathway for single-strand DNA break repair and maintenance of genetic stability. It was also demonstrated to co-localize with aprataxin, PARP-1, and p53 on chromatin [65]. The *HRAS* proto-oncogene variation is extremely rare and classified as a pathogenic variant according to ACMG, ClinVar, and UniProt classification. In a study that demonstrated the variety in the molecular profile of 96 surgically resected thyroid nodules, RAS (*HRAS, NRAS, KRAS*) oncogene mutations were present, either alone or with other mutations in almost one-half of cases (46 of 96 cases; 48%) [57].

In summary, *ARHGEF28* could harbor the germline mutation predisposing to PTC. It may affect two pathways: the p53 pathway by binding to SQSTM1, and the regulation of *FOXE1* expression by binding to MYH9. The additional second hits in the *HRAS* proto-oncogene and *XRCC1*, which is essential for DNA break repair and maintenance of genetic stability, could lead to PTC progression. Additional studies on the molecular function of the genes suggested by our research, mainly the *ARGHEF28* gene, will be beneficial for understanding the molecular mechanism of DTC.

## 4. Patients and Methods

### 4.1. Family Member Evaluation

The local ethical committee approved the current research on the 26th of January 2016 (approval number 0185-15-SOR). Many patients from the same family, followed by the endocrine unit of SUMC, have thyroid pathology (mainly DTC and MNG). Thus, we invited all family members to participate in this study. All participants were evaluated phenotypically by an endocrinologist. All computerized data of family members included in the study (patients followed for DTC and/or MNG and those with anatomically and functionally normal thyroid glands) were available for the clinicians of the research team. The clinical evaluation included medical history, physical examination, a routine neck US according to the current accepted clinical guidelines [3], cytological results whenever FNA was conducted, histopathological results for those who were operated on, and follow-up data when appropriate. The biochemical evaluation included a thyroid function test, thyroglobulin concentration, and the presence of anti-thyroglobulin and anti-thyroid-peroxidase antibodies.

### 4.2. Exome Capture and Sequencing

gDNA was extracted from participants’ blood and submitted to a commercial company for exome capture and sequencing. Three companies were used: (1) Otogenetics Corporation (Norcross, GA, USA). Illumina libraries were made from qualified fragmented gDNA using NEBNext reagents (New England Biolabs, Ipswich, MA, USA, catalog# E6040), and the resulting libraries were subjected to exome enrichment using NimbleGen SeqCap EZ Human Exome Library v2.0 (Roche NimbleGen, Inc., Madison, WI, USA, catalog# 05860482001) following manufacturer’s instructions. Enriched libraries were tested for enrichment by qPCR and size distribution and concentration by an Agilent Bioanalyzer 2100. The samples were then sequenced on an Illumina HiSeq2000. The average depth of sequence was X30. Data were analyzed for quality, exome coverage, and exome-wide SNP/InDel using the platform provided by DNAnexus (DNAnexus, Inc, Mountain View, CA, USA). (2) Theragene. SureSelect XT Human all exon V6 kit was used for library preparation, the target size was 58 Mb, and the coverage uniformity at X10 coverage was ≥90%. Results were analyzed using QIAGEN’s Ingenuity Variant Analysis software (www.qiagen.com/ingenuity, QIAGEN Redwood City). The depth of the sequences was X30, X50, and X100. (3) Macrogen Humanizing Genomics. Agilent SureSelect V5 was used for library preparation, and the depth of the sequences was X50.

For the DNA purified from the formalin-fixed, paraffin-embedded (FFPE) tissue, WES analysis was carried out at Theragene using Illumina NovaSeq 6000 sequencing system. SureSelect XT Human all exon V6 kit was used for library preparation, the target size was 58 Mb, and the coverage uniformity at X10 coverage was ≥90%. Results were analyzed using QIAGEN’s Ingenuity Variant Analysis software (www.qiagen.com/ingenuity, accessed on 1 July 2020, QIAGEN Redwood City).

### 4.3. DNA Purification from the FFPE Tissue

DNA from a patient’s extracted thyroid cancer preserved in formalin was purified using TAIGEN LabTurbo Handbook kit version 2.4 D (https://labturbo.com/wp-content/uploads/2021/08/LabTurbo-Kit-Handbook-48C_2020-10-17_ENCE.pdf, accessed on 10 June 2020).

### 4.4. Genetic Analysis

gDNA was extracted from the blood of 17 individuals (Figure 1), most of them from the third generation. For exome capture and sequencing, gDNA of patients III8, XSIII5, and IVA5 were submitted to Otogenetics Corporation, III10 and XFIII5 to Theragene, and XFIII4 to Macrogen (Figure 1). Furthermore, DNA of patient III8 was also extracted from his FFPE thyroid cancer tissue and submitted to Theragene corporation. The results were analyzed using the Fabric Genomics, Franklin, and Ingenuity pipelines. ClinVar, UniProt, American College of Medical Genetics (ACMG) criteria, SIFT, CADD, PolyPhen, splice AI, and dbscSNV Ada tools were used to evaluate the pathogenicity of the variants. The public databases used to exclude variants based on frequency in the general population were gnomAD browser, 1000 genomes, EVS, and ExAC. Our in-house Bedouin exome sequence database, created while looking for variations in this population, excluding thyroid patients, included 780 sequences. An additional database for healthy Bedouins [49] and more than 1000 exomes of Qatari genomes whose majority is of the Bedouin population [50] were used to exclude frequent variants. The Biological General Repository for Interaction Datasets (BioGRID) was used to find genetic and protein interaction data for the susceptible variations.

### 4.5. Verification of the Variations

The primers used in this research for PCR amplification of the different genes’ loci are detailed in Appendix A. Direct sequencing of the PCR products was performed as detailed in [66].

## Figures and Tables

**Figure 1 ijms-24-08233-f001:**
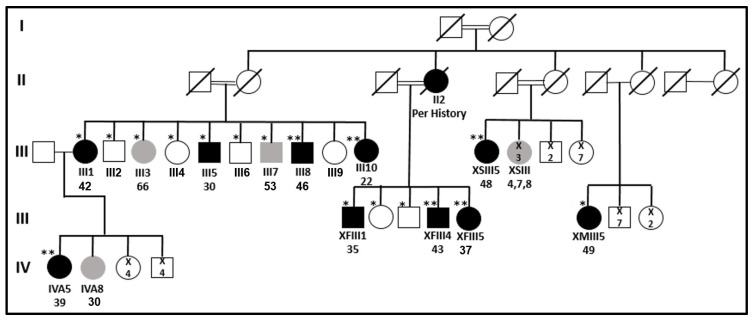
**Simplified pedigree of the Bedouin kindred.** The two left branches are our current focus. Black symbols denote FNMTC; grey symbols denote MNG. * Available DNA, ** whole exome sequence available. 
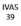
: top—patient’s code; bottom—patient’s age at diagnosis. 
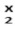
: number of individuals.

**Figure 2 ijms-24-08233-f002:**
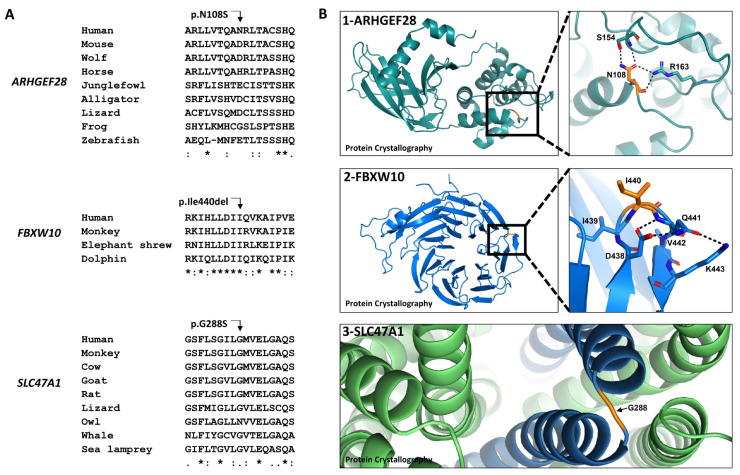
Evolutionary conservation and 3D structure of ARHGEF28, FBXW10, and SLC47A1 proteins. (**A**) The evolutionary conservation in the region of the proteins containing the variations. An * (asterisk) indicates a fully conserved residue. A : (colon) indicates conservation between groups of strongly similar properties. A . (period) indicates conservation between groups of weakly similar properties. (**B**) The predicted 3D structure of the proteins. Part 1—ARHGEF28, the left side represents residues 1–285 of the overall 3D structure of the ARHGEF28 protein, based on the alpha fold model (accession code—Q8N1W1). The right side presents a zoomed-in view of the site of the affected amino acid, Asn108, and its hydrogen bonds (dashed lines) connecting it to the Ser154 and Arg163 amino acids. Part 2—FBXW10, the left side represents the propellor domain, residues 339–693 of the overall 3D structure of the FBXW10 protein, based on the alpha fold model (accession code—Q5XX13). The right side presents a zoomed-in view of the site of the affected amino acid, Ile440, and its hydrogen bonds connecting it to the Asp438 and Lys443 amino acids. Part 3—SLC47A1, zoomed-in view of the site of the affected amino acid, Gly288 of the SLC47A1 protein, based on the alpha fold model (accession code—Q96FL8).

**Table 1 ijms-24-08233-t001:** Clinical characteristics of all family members of the two branches of main interest.

ID	Gender M/F	Current Age	Age at Diagnosis	Pathology or US Results ^¥^	Largest Diameter	Variant ^#^	Multifocality ^#^	LN MTS ^#^	RAI ^#^	Status (Year) ^§^
III1	F	79	42	PTC	NA	NA	NA	NA	NA	Free of disease (2016)
III2	M	76	69	NT						NT (2016)
III3	F	73	66	Benign MNG ˄	2 cm					Benign MNG (2016)
III4	F	70	63	NT						NT (2016)
III5	M	63	30	PTC	NA	NA	NA	Yes	Yes	Lost to follow-up
III6	M	61	54	NT						NT (2016)
III7	M	60	53	Benign MNG ~	4 cm					Benign MNG (2016)
III8	M	57	46	PTC	1.6 cm	Follicular	Yes, 5 foci	No	100	Free of disease (2021)
III9	F	54	NA							
III10	F	50	22	PTC	2.5 cm	Follicular	Yes, 3 foci	No	Yes- Twice	Free of disease (2021)
XFIII1	M	58	35	PTC	1.5 cm	Follicular	Yes, 3 foci	Yes	250 *	Lost to follow-up
XFIII2	F	NA								
XFIII3	M	NA								
XFIII4	M	51	43	PTC	2.1 cm	Classic/follicular	Yes, “many foci”	Yes	100	Free of disease (2022)
XFIII5	F	50	37	PTC	NA	NA	NA	Yes	150	Free of disease (2022)
IVA5 ^@^	F	53	39	PTC	1.4 cm	Classic	Yes, at least 4 foci	Yes	150	Free of disease (2011) ^@^
IVA8	F	45	30	Benign MNG	3 cm					Lost to follow-up

At the time of diagnosis or clinical evaluation, ^¥^ papillary thyroid carcinoma, multinodular goiter, or normal thyroid by ultrasound, ^#^ for family members with PTC, ^§^ last follow-up (year of last visit), ˄ benign ultrasound features—there was no indication for fine-needle aspiration biopsy, ~ fine needle aspiration reported benign nodule(s), * in 2 separate doses, ^@^ at 12/2020 (when she was 50 years old) IVA5 was diagnosed with adenocarcinoma of the lung. Abbreviations: LN MTS—lymph node metastasis; RAI—treatment with radioactive iodine (activity in millicurie if the information was available); F—female; M—male; PTC—papillary thyroid carcinoma; NA—not available (patient refused clinical and genetic evaluation); NT—normal thyroid; MNG—multinodular goiter. The family members’ annotations are according to Figure 1.

**Table 2 ijms-24-08233-t002:** Evaluating and prioritizing potential candidate variants in patients III8, III10, IVA5, XFIII4, and XFIII5 based on several criteria.

Number of Variants after Filtration	Filtration Criteria for Homozygous or Heterozygous Variations
644,298 → 12,626	Shared variations between the WESs of patients III8, III10, IVA5, XFIII4, and XFIII5
12,626 → 84	Presence in the general databases—1KG, EVS, ExAC, and gnomAD—at frequencies < 1%
84 → 54	Presence in our internal laboratory exome database of the Bedouin population at frequencies < 1%
54 → 4	Familial segregation analysis (detailed in Appendix A)

**Table 3 ijms-24-08233-t003:** Familial segregation analysis for the four potential candidate variants compatible with the association of a causative variant.

Gene Symbol and Position	Ref/Alt	III1 A	III2 H	III4 H	III5 A	III6 H	III8 A	III10 A	IVA5 A	XFIII1 A	XFIII4 A	XFIII5 A
*ARHGEF28* chr5:73048875	A > G	−/+	+/+	+/+	−/+	−/+	−/+	−/+	−/+	−/+	−/+	−/−
*FBXW10* chr17:18661699	delCAT	−/+	−/+	+/+	−/+	+/+	−/+	−/+	−/+	−/+	−/+	−/+
*SLC24A4* chr14:92953131	A > G	−/+	+/+	−/+	−/+	+/+	−/+	−/+	−/+	−/+	−/+	−/+
*SLC47A1* chr17:19459316	G > A	−/+	−/+	+/+	−/−	+/+	−/+	−/+	−/+	−/+	−/+	−/+

The family members’ annotations are according to Figure 1. “A” represents affected PTC individuals, and “H” represents healthy individuals. Underlined font: incomplete penetrance. −/−: homozygote for the variation; −/+: heterozygote; +/+: normal. Positions on chromosomes are according to GRCh37/hg19.

**Table 4 ijms-24-08233-t004:** Expression value, frequency in the population of origin, and predictions for damage analysis of the four potential candidate variants compatible with the association of a causative variant.

Gene Symbol Position Ref/Alt	cDNA Protein	GTEx (TPM) ^@^	Prevalence in the Population of Origin						Prediction Tools			ACMG	ClinVar
			Internal Lab Frequency *		Saudi’s Frequency ^&^		Qatari’s Allele Number ^$^						
			Het.	Hom.	Het.	Hom.	Het.	Hom.	SIFT	CADD	PolyPhen		
*ARHGEF28* chr5:73048875 A > G	c.323A > G p.Asn108Ser	19.44	0.0057	0.0006	0	0	0.003	0	0.14 (tolerated)	10.71	0.02 (benign)	BS2, BP4 (likely benign)	VUS
*FBXW10* chr17:18661699 delCAT	c.1317_ 1319delCAT p.Ile440del	1.22	0.0128	0	0	0	0	0	-	-	-	PM2, PM4 (VUS)	-
*SLC24A4* chr14:92953131 A > G	c.1532A > G -	0.20	-	-	0	0	0.0065	0	-	4.68	-	BS2, BP4, BP6 (benign)	Benign
*SLC47A1* chr17:19459316 G > A	c.862G > A p.Gly288Ser	7.45	0.0147	0.0006	0.026	0.013	0.012	0	0.83 (tolerated)	22	0.73 (possibly damaging)	BP4, PM2 (VUS)	-

The family members’ annotations are according to Figure 1. ^@^ Expression value according to GTEx portal. * Variant frequency in our collection of 780 individual Bedouin exomes (without any thyroid disorders). ^&^ Variant frequency in Saudi’s Bedouin database of 77 exomes [49]. ^$^ Variant frequency in Qatari genome of ~1000 exomes, with a majority of the Bedouin population [50]. Het, heterozygote variant. Hom, homozygote variant. VUS, a variant of uncertain significance. Positions on chromosomes are according to GRCh37/hg19.

**Table 5 ijms-24-08233-t005:** Evaluating and prioritizing potential candidate variants in the malignant tissue of patient III8 based on several criteria.

Number of Variants after Filtration	Filtration Criteria
217,476 → 6697	Variations that exist in the WES of the malignant thyroid tissue and are absent in the gDNA of the same patient—III8
6697 → 1527	Presence in the general databases—1KG, EVS, ExAC and gnomAD—at frequencies < 5%
1527 → 2	Applying a case panel that presents only variations in genes associated with PTC—see Appendix A and list of 13 somatic genes [57]

## Data Availability

The data will be made available upon request. Web sources: GTEx portal: https://gtexportal.org/home/gene/ARHGEF28, accessed on 7 September 2022; BioGRID Database: https://thebiogrid.org/122127/summary/homo-sapiens/arhgef28.html, accessed on 20 March 2023.

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
