# Peer review of "Novel Susceptibility Genes Drive Familial Non-Medullary Thyroid Cancer in a Large Consanguineous Kindred"

_ijms, 2023, doi:10.3390/ijms24098233_

Round 1
Reviewer 1 Report
Manuscript Title: Novel Susceptibility Genes Drive Familial Non-Medullary 2 Thyroid Cancer in a Large Consanguineous Kindred
. . . . . . . . . . . . . . . . . . . . . . . . . . . . . . . . . . . . . . . . . . . . . . . . . . . . . . . . . . . . . . . . . . . . . . . . . . . . . . . . . . . . . .
Comments on the current macula on New Susceptibility Genes Driving a Major Relative Intra-Medullary Thyroid Cancer are presented below.
It would be appropriate to mention the date of the approval document obtained from the local ethics committee in the text.
What was used as statistical analysis?
. . . . . . . . . . . . . . . . . . . . . . . . . . . . . . . . . . . . . . . . . . . . . . . . . . . . . . . . . . . . . . . . . . . . . . . . . . . . . . . . . . . . . .
Reviewer 2 Report
Familial non-medullary thyroid cancer (FNMTC) is a well-differentiated thyroid cancer (DTC) of follicular cell origin in two or more first-degree relatives.
In this study, a Whole Exome Sequence (WES) analysis of six family patients was performed, to identify the variation/s causing FNMTC in an extended consanguineous family consisting of 16 papillary thyroid carcinoma (PTC) cases. An association between ARHGEF28, FBXW10, and SLC47A1 genes with FNMTC was shown. ARHGEF28 could harbor the germline mutation predisposing to PTC. It may affect two pathways- the p53 pathway by binding to SQSTM1, and the regulation of FOXE1 expression by binding to MYH9. The additional second hits in the HRAS proto-oncogene and XRCC1 could lead to PTC progression.
The Authors concluded that the predisposition by the germline variations and a second hit by somatic variations could lead to the progression to PTC.
The study is interesting. Figures and tables are clear.
Since only about 28% of the cited references have been published after the year 2018, I suggest only to cite and discuss more recently published papers, in particular in the Background section, about different types of thyroid cancer, in order to focus better the argument, such as 10.1016/j.semcancer.2020.11.013
Reviewer 3 Report
The manuscript by Majdalani et al identified novel hotspot genes associated with thyroid cancers in the Bedouin kindred. The manuscript was written in a very clear and logic fashion, the results are presented in very well organized tables. However, a few points need to be addressed before the acceptance of this manuscript.
1. In figure 2A, the authors need to give positional information of the amino acid sequences in the alignment. And this will make figure 2A more relevant to figure 2B.
2. It would be nice if the authors could put the PPI analysis results in the supplements.
